# Theoretical and Experimental Analysis on Influence of Natural Airflow on Spent Fuel Heat Removal in Dry Cask Storage

**Ratiko Ratiko** [1,*], **Raden Sumarbagiono** [1], **Aisyah Aisyah** [1], **Wati Wati** [1], **Kuat Heriyanto** [1], **Mirawaty Mirawaty** [1], **Pungky Ayu Artiani** [1], **Yuli Purwanto** [1], **Dwi Luhur Ibnu Saputra** [1], **Jaka Rachmadetin** [1], **Risdiyana Setiawan** [1], **Arifin Istavara** [1] and **Abdullah Ahmad Rauf** [2]

1   Research Center for Radioactive Waste Technology (PRTLR), Nuclear Power Research Organization (ORTN), National Research and Innovation Agency (BRIN), Tangerang Selatan 15310, Indonesia; raden.sumarbagiono@brin.go.id (R.S.); aisyah@batan.go.id (A.A.); wati@batan.go.id (W.W.); kuat@batan.go.id (K.H.); mirawaty@batan.go.id (M.M.); pungky@batan.go.id (P.A.A.); yuli_p@batan.go.id (Y.P.); Ibnu.saputra@batan.go.id (D.L.I.S.); jaka@batan.go.id (J.R.); risdiyana@batan.go.id (R.S.); istavara@batan.go.id (A.I.)
2   Department of Mechanical Engineering, Politeknik Negeri Jakarta, Depok 16424, Indonesia; ahm81r@gmail.com
*   Correspondence: ratiko@batan.go.id

**Abstract:** A key issue contributing to the success of NPP technology is the safe handling of radioactive waste, particularly spent nuclear fuel. According to the IAEA safety standard, the spent fuel must be stored in interim wet storage for several years so the radiation and the decay heat of the spent fuel will decrease to the safe limit values, after which the spent fuel can be moved to dry storage. In this study, we performed a theoretical analysis of heat removal by natural convection airflow in spent nuclear fuel dry storage. The temperature difference between the air inside and outside dry storage produces an air density difference. The air density difference causes a pressure difference, which then generates natural airflow. The result of the theoretical analysis was validated with simulation software and experimental investigation using a reduced-scale dry storage prototype. The dry storage prototype consisted of a dry cask body and two canisters stacked to store materials testing reactor (MTR) spent fuel, which generates decay heat. The cask body had four air inlet vents on the bottom and four air outlet vents at the top. To simulate the decay heat from the spent fuel in the two canisters, the canisters were wrapped with an electric wire heater that was connected to a voltage regulator to adjust the heat power. The theoretical analysis results of this study are relatively consistent with the experimental results, with the mean relative deviation (*MRD*) values for the prediction of air velocity, the heat rate using natural airflow, and the heat rate using the thermal resistance network equation are +0.76, −23.69, and −29.54%, respectively.

**Keywords:** spent fuel; dry cask storage; natural airflow; heat removal

## 1. Introduction

A central issue of the nuclear power plant (NPP) program is the generation of radioactive waste. Thus, the safe handling and management of radioactive waste is very important. The most high-risk and dangerous radioactive waste is spent nuclear fuel because of the risk of a self-sustaining chain reaction (criticality), the very high radioactivity, and the high decay heat.

There are several studies on the handling and storage of spent fuel [1–10]. After remaining in interim wet storage for more than 4 years, spent fuel can be moved to dry cask storage.

An important research topic related to spent fuel dry storage is the removal of spent fuel decay heat. The dry storage must be designed in such a way that an optimal heat flow is obtained so the temperature of the spent fuel in dry storage does not exceed the

maximum specified limit. The maximum temperature limit for spent fuel stored in dry storage depends on the type of spent fuel. In general, after a long storage period (tens or even hundreds of years) and at a lower temperature, the spent fuel is expected to be safer, more durable, and more resistant to material aging processes, such as corrosion, so that the radionuclides stored in the spent fuel will not escape through the spent fuel cladding. One method of increasing the heat removal from concrete dry storage is adding an air gap as a ventilation channel in the dry storage

There are many studies on spent fuel heat removal from non-ventilated dry storage and ventilated dry storage. Several studies on heat removal from non-ventilated dry storage, among others, have discussed decay heat removal after a long-term storage period using the ALGOR code [11,12], heat transfer enhancement in non-ventilated dry storage using inert gas [13], and minimizing the deviation of the thermal load in a non-ventilated dry cask [14].

Regarding ventilated dry storage, there are more studies that have discussed heat removal [15–18]. Several studies have investigated thermal modeling of the natural convection using the ANSYS Fluent code. One analysis showed that natural convection has an influence on the peak cladding temperature (PCT) and the canister surface temperature (CST) [16]. In a similar study, the heat removal performance of a ventilated concrete storage cask was evaluated under both normal and accident conditions to meet the safety standard for the storage of spent fuel. [17]. Numerical modeling was also used in ventilated dry storage using computational fluid dynamics (CFD) to investigate several inner circulations in the cask at the flow channel. These circulations had an influence on the heat dissipation of the dry storage [15]. In another study, a comparison between numerical/mathematical modeling and experiments on temperatures at the inlet and outlet vents was conducted on ventilated dry storage in the Zaporizhska NPP [18].

In summary, there has been enough research on heat removal from ventilated dry storage; however, these studies have focused more on the analysis and simulation of the heat transfer and temperatures. In our study, we conducted an analysis of the numerical prediction of the natural airflow in the air gap of ventilated dry storage, which is utilized to cool the spent fuel; the airflow prediction was then compared with the experimental data. We discuss several equations that can be used to predict the flow rate or air velocity in a ventilated dry storage prototype. The prediction of air velocity was then compared with the experimental data. The airflow was used to calculate the predicted heat removal rate and then compared with the heat calculation using the thermal resistance network equation. It was then compared with the heat value from the experiment. In this study, the spent fuel stored in dry storage was from our research reactor (material testing reactor—MTR), with a fuel temperature limit lower than 150 °C [19,20].

## 2. Methodology

### 2.1. Experimental

To validate the calculation and numerical modeling, an experimental investigation was performed using a reduced-scale dry storage cask prototype for MTR spent fuel with an outer diameter of 750 mm and an inner diameter of 390 mm. We placed it in an air-conditioned laboratory room, as shown in Figure 1a,b. This dry cask was 1.37 m in height and had an air gap through 4 inlet vents at the bottom and 4 outlet vents at the top. These eight vents had a cross-sectional area of 17,500 mm$^2$. The vents could be closed and opened. If the vents were open, the dry storage prototype was ventilated dry storage. The air could flow into the dry storage through the vents by natural convection and cool the outer walls of the canister surface. If the vents were closed, then the prototype was non-ventilated dry storage; however, the conduction material in this case was air.

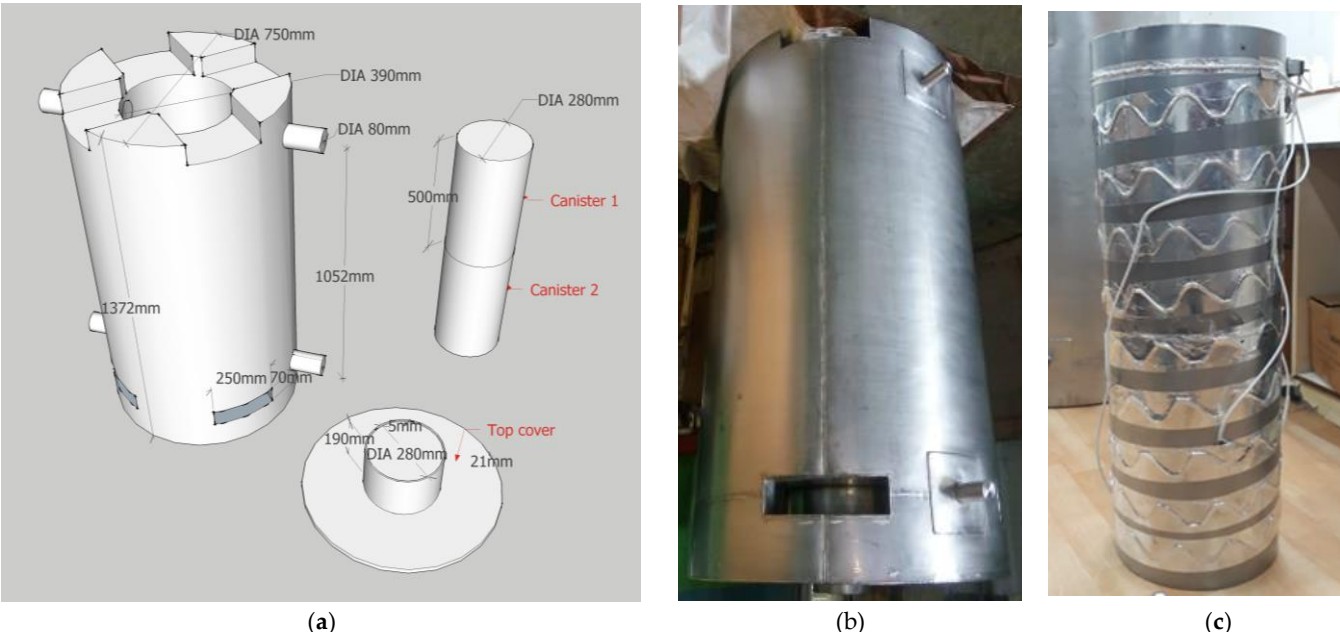

**Figure 1.** The dry cask prototype. (**a**) design; (**b**) dry cask wall; (**c**) canister.

The dry cask had two stainless steel canisters that were vertically stacked. To simulate the spent fuel decay heat, the surfaces of the two canisters were wrapped with an electric heater wire (nickel cable), as shown in Figure 1c, which was connected to a voltage regulator so that the cable heat power could be adjusted to be equal or similar to the decay heat power generated by the spent fuel.

The temperature of the dry cask, the temperature of the air gap, and the airflow through the air gap were measured in this study. The temperatures were measured using a thermocouple, which had ±1.5 °C measurement accuracy for the investigated temperature range and was placed at 8 different measurement points, as shown in Figure 2.The National Instruments (NI) temperature input module was used to take measurements from the thermocouples. The NI data acquisition (DAQ) measurement hardware was connected to a computer with LabVIEW software (v. 2018 SP1, NI, Austin, TX, USA) for viewing, analyzing, and logging the measurement data. Temperature measurements were carried out every 5 min for about 8 h for each variation of the voltage value from the heater, which was regulated by a voltage regulator (50, 75, 100, 125, and 150 V). In addition, temperature measurements were also carried out on several variations of the vents in closed and half-closed conditions.

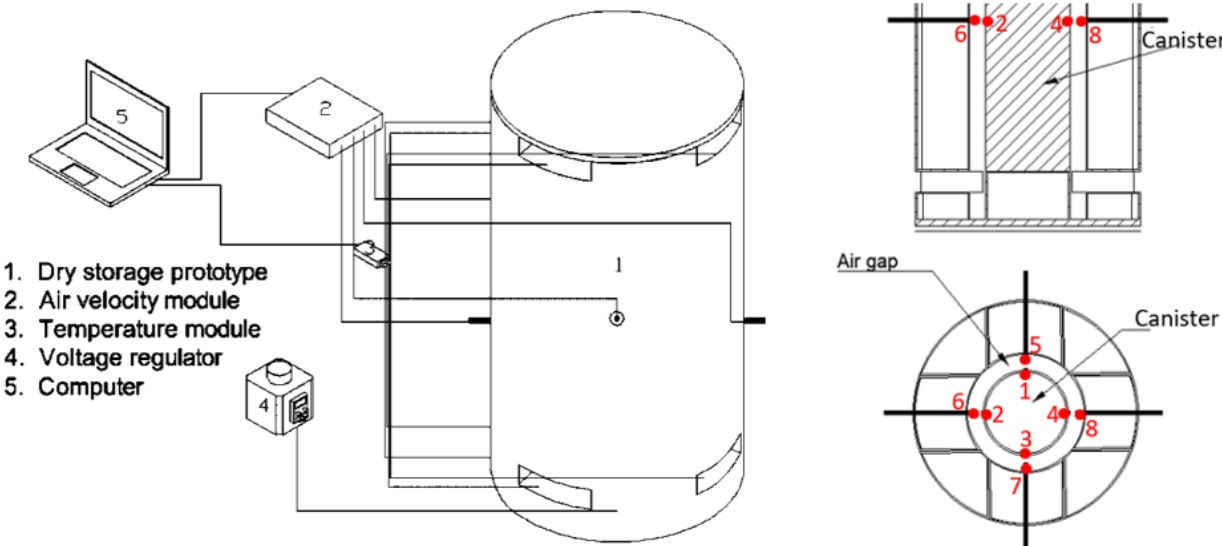

**Figure 2.** Experimental set-up.

The airflow measurement was conducted using a Dantec TC anemometer, a hotwire thermal anemometer with a measurement accuracy of ±0.06 m/s for the velocity range investigated in this study. This air velocity meter can measure very low air velocity. The air velocity measured in this study was less than 1 m/s. The measurement of air velocity in the air gap was carried out using the same method as the temperature measurement. In addition, the air velocity measurement of the outlet vent temperature (from 30 to 80 °C) was also carried out. The outlet vent temperature was adjusted by increasing the heater voltage from 50 to 200 V using the voltage regulator.

The overall experimental set-up is shown in Figure 2.

*2.2. Model*

The analysis of numerical modeling in this study was calculated using heat transfer and airflow analysis.

The natural airflow modeling and the heat transfer analysis were conducted using CONTAM (v. 3.1, NIST, Gaithersburg, MD, USA). CONTAM is a multizone indoor air quality and ventilation analysis software that can determine airflows driven by mechanical, wind, and buoyancy effects.

In this study, the natural convective heat transfer occurred on the outer surface of the canister through the air gap inside the dry cask according to the following common equation:

$$\dot{Q}_{conv} = hA_c\left(T_c - T_{ag}\right) \tag{1}$$

where $h$ is the convective heat transfer coefficient in $\frac{W}{m^2 K}$, $A_c$ is the outer surface area of the canister in $m^2$, $T_c$ is the temperature at the outer surface of the canister in K, and $T_{ag}$ is the temperature at the air gap in K.

For a simpler understanding of the heat transfer process, the thermal resistance concept can be used in a steady state condition.

The heat thermal equations above are analogous to an electric current $I$ in a simple equation, as follows:

$$I = \frac{V_1 - V_2}{R_e} \tag{2}$$

where $I$ is the electric current in $A$, $V$ is the voltage in V, and $R_e$ is the electric resistance in $\Omega$ [21].

This analogy is described in Figure 3a,b by the following equation:

$$\dot{Q} = \frac{T_1 - T_2}{R} \tag{3}$$

where $T$ (temperature in K) is analogous to $V$ (voltage) and $R$ (thermal resistance in K/W) is analogous to $R_e$ (electric resistance).

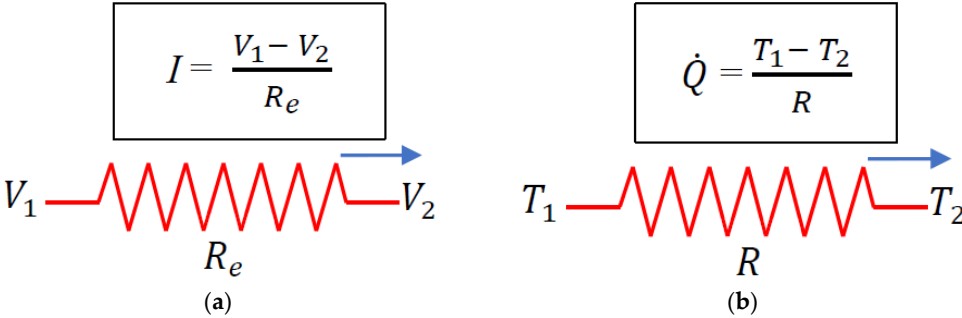

**Figure 3.** Thermal resistance concept compared to the electrical resistance concept. (**a**) Electric current flow; (**b**) heat flow.

Using Equation (4), Equation (1) can be rearranged as

$$\dot{Q}_{conv} = \frac{T_c - T_{ag}}{R_{conv}} \tag{4}$$

where the convection resistance is

$$R_{conv} = \frac{1}{h_{conv} \, A_c} \tag{5}$$

The heat transfer from the outer surface of the canister to the ambient air gap consists of several layers of different materials. The heat rate at steady state can be expressed as follows (Figure 4) [21,22]:

$$\dot{Q} = \frac{T_1 - T_\infty}{R_{total}} \tag{6}$$

where $R_{total}$ is the total thermal resistance and is expressed as

$$R_{total} = R_{air\ gap} + R_{cask\ wall} + R_{conv} = \frac{1}{h_1 \, A_1} + \frac{\ln\left(\frac{r3}{r2}\right)}{2\pi \, h_{cask} \, k_{cask}} + \frac{1}{h_3 \, A_3} \tag{7}$$

where $R_{cask\ wall}$ is the thermal conduction resistance from the cask wall in °C/W, $h$ is the height of the cask wall in m, and $k_{cask}$ is the thermal conductivity of the cask wall in W/mK.

In cylinders and spheres, the heat conduction area ($A$) is not constant. $A$ depends on $r$, it varies in the direction of the heat transfer, and thus, can be calculated as [22]

$$\int_{r2}^{r3} \frac{\dot{Q}_{cond,\ cyl}}{A} \, dr = -\int_{T2}^{T3} k \, dT \tag{8}$$

Substituting $A = 2\pi rh$ and executing the integration gives

$$\dot{Q}_{cond,\ cyl} = 2\pi \, h \, k \, \frac{T_2 - T_3}{\ln\left(\frac{r3}{r2}\right)} \tag{9}$$

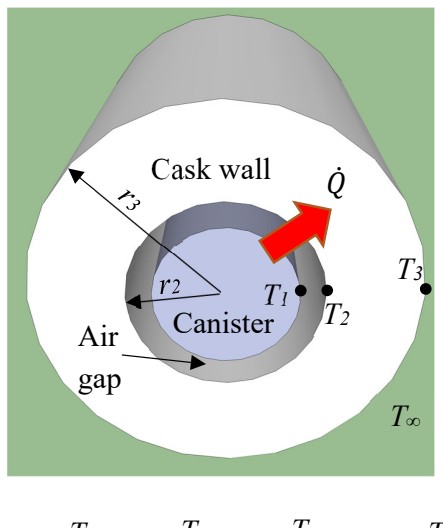

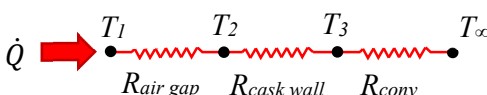

**Figure 4.** The thermal resistance network for heat transfer in the dry storage prototype.

where $h_1$ and $h_3$ are the natural convection heat transfer coefficients in W/m$^2$K. The empirical correlations for the average Nusselt number for natural convection in a vertical cylinder (dry cask) are [13,16,22]

$$Nu = \frac{hL_c}{k}, \tag{10}$$

$$Nu = 0.59 \, Ra^{1/4} \text{ for } 10^4 < Ra < 10^9, \tag{11}$$

$$Nu = 0.1 \, Ra^{1/3} \text{ for } 10^9 < Ra < 10^{13}, \tag{12}$$

or

$$Nu = (0.825 + \frac{0.387 \, Ra^{1/6}}{\left(1 + (0.492/Pr)^{9/16}\right)^{8/27}})^2 \text{ for entire range of } Ra \tag{13}$$

where

$$Ra = Gr \, Pr \tag{14}$$

and

$$Gr = \frac{g \, \beta \, (T_s - T_\infty) \, L^3}{v^2}, \tag{15}$$

$$Pr = \frac{C_p \, \mu}{k} \tag{16}$$

Another method of calculating the heat transfer in this dry storage prototype is to use the equation of natural airflow from the study conducted by Ratiko et al. [23]. The air density difference between the input and output leads to a pressure difference, which then generates natural airflow.

The pressure difference can be calculated using the hydrostatic equation, as follows:

$$\Delta P = \Delta \rho g z \tag{17}$$

where $\Delta P$ is the pressure difference in Pa, $\Delta \rho$ is the air density difference in kg/m$^3$, $g$ is gravitational acceleration in m/s$^2$, and $z$ is the elevation (relative to the bottom of the vessel) in m. $\Delta P$ is generated because of the temperature difference between the two adjacent spaces or rooms.

The airflow that enters or leaves the dry storage through the air gap can be calculated using the following equation [24–26]:

$$F = C_d A \sqrt{\frac{2\Delta P}{\rho}} \qquad (18)$$

where $F$ is the volume airflow rate in m$^3$/s, $C_d$ is a dimensionless discharge coefficient for the inlet or outlet vent. The $C_d$ value depends on the geometry of the orifice, and for a small vent or opening, it depends strongly on the Reynolds number. In this study, the vent dimension was 250 mm $\times$ 70 mm and categorized as a small square-edged opening. For this type of opening, the typical $C_d$ value is 0.6. $A$ is the inlet or outlet cross-sectional area in m$^2$.

The pressure difference in the air gap, as shown in Figure 5, is as follows:

$$\Delta P_i = \frac{1}{2}(z_1 + z_2) g (\rho_i - \rho_o) \qquad (19)$$

where $\Delta P_i$ is the residual pressure of the dry cask in Pa, $\rho_i$ and $\rho_o$ are the air densities inside and outside the air gap, respectively, in kg/m$^3$, $z_1$ is the elevation from the bottom of the dry cask to the inlet midpoint in m, $z_2$ is the elevation from the bottom of the dry cask to the outlet midpoint in m.

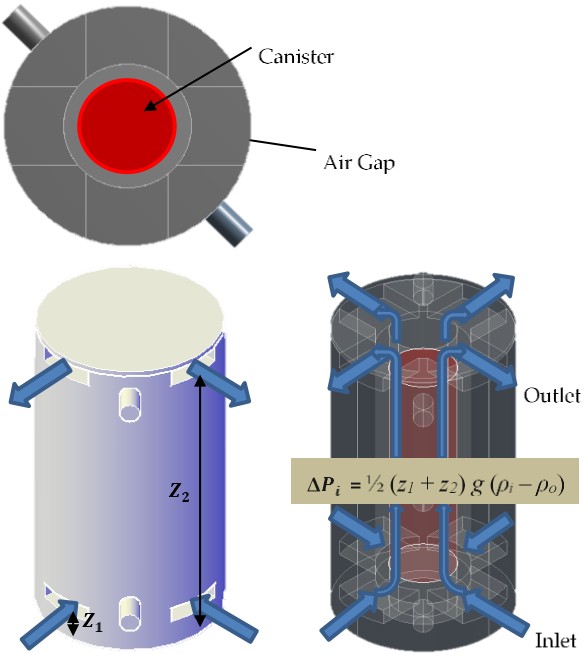

**Figure 5.** Pressure difference and natural airflow that enters and leaves the dry cask.

Using the equation in Ratiko et al.'s study [23], the airflow that enters and leaves the dry storage can be calculated as follows:

$$F = C_d A \sqrt{(z_2 - z_1) g \frac{(T_i - T_o)}{T_i}}, \qquad (20)$$

while the velocity can be calculated as

$$v = C_d \sqrt{(z_2 - z_1) g \frac{(T_i - T_o)}{T_i}} \qquad (21)$$

where $T_i$ is the air temperature inside the dry cask and $T_o$ is the outside air temperature, both in Kelvin.

The natural convection airflow in the dry storage prototype that enters and leaves the dry storage vents through the air gap is presented in Figure 5.

## 3. Results and Discussion

### 3.1. Experimental Data Results

To validate the calculation and numerical modeling, the temperature and airflow through the air gap were measured. The temperature measurement was taken at several points on the outer surface of the canister, in the air gap, in the outlet vents, and in the air-conditioned laboratory room where the dry storage prototype was placed. From the temperature measurements at several heater voltages, two measurement data points at 100 and 125 V voltages are shown in Figures 6 and 7, respectively.

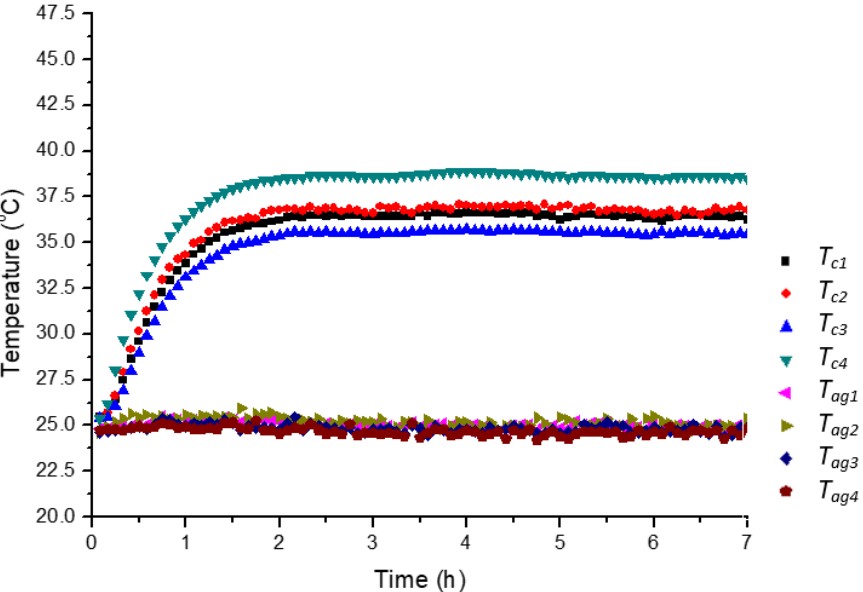

**Figure 6.** Temperature measurement at a regulator voltage of 100 V.

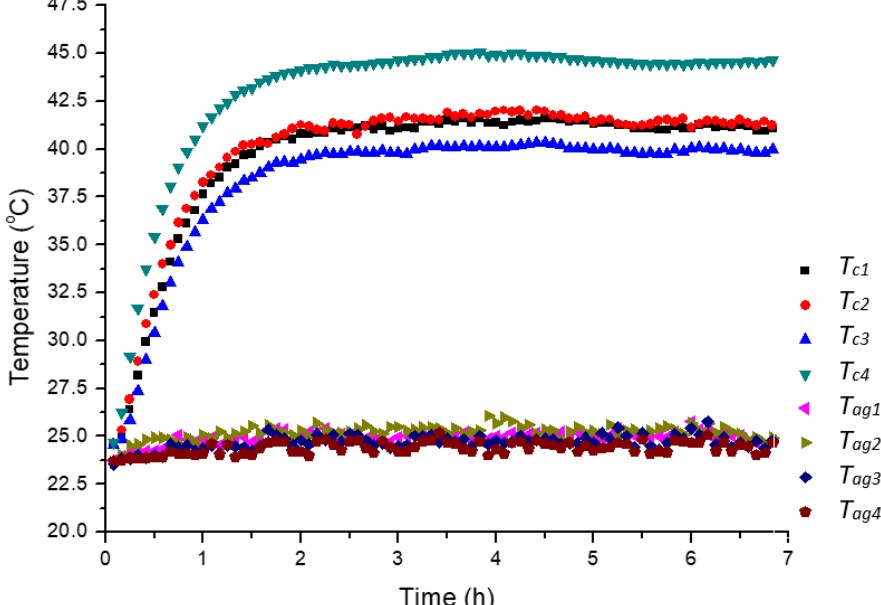

**Figure 7.** Temperature measurement at a regulator voltage of 125 V.

The temperature measurements were performed at several different heat rates of the electric heater wire wrapped around the canister. The heater temperature was adjusted using a variable voltage regulator.

The temperature measurement results at 100 V regulator voltages are shown in Figure 6. The canister surface temperatures at four measurement points ($T_{c1}$–$T_{c4}$) increased with a function of time. At steady state conditions, the average canister temperature of $T_{c1}$, $T_{c2}$, $T_{c3}$, and $T_{c4}$ was 36.88 °C. The temperatures in the air gap ($T_{ag1}$–$T_{ag4}$) also increased with a function of time, but the increase was much smaller than $T_c$. The average air gap temperature at steady state conditions was 24.8 °C, slightly higher than the laboratory room temperature of 23.3 °C.

When the heat rate of the canister was increased (by increasing the voltage on the voltage regulator), the canister temperature and the air gap temperature also increased. Figure 7 presents the temperature data at a regulator voltage of 12 V. The average temperature of the canister was 41.82 °C, and the average temperature of the air gap was 24.9 °C.

Additional experimental data needed for analysis in this study were the temperature comparisons of the eight vents that were closed versus those that were open. Figure 8 shows the comparison between the temperatures on the canister surface and in the air gap at regulator voltages of 100 and 125 V when the eight vents were opened compared to when the vents were closed.

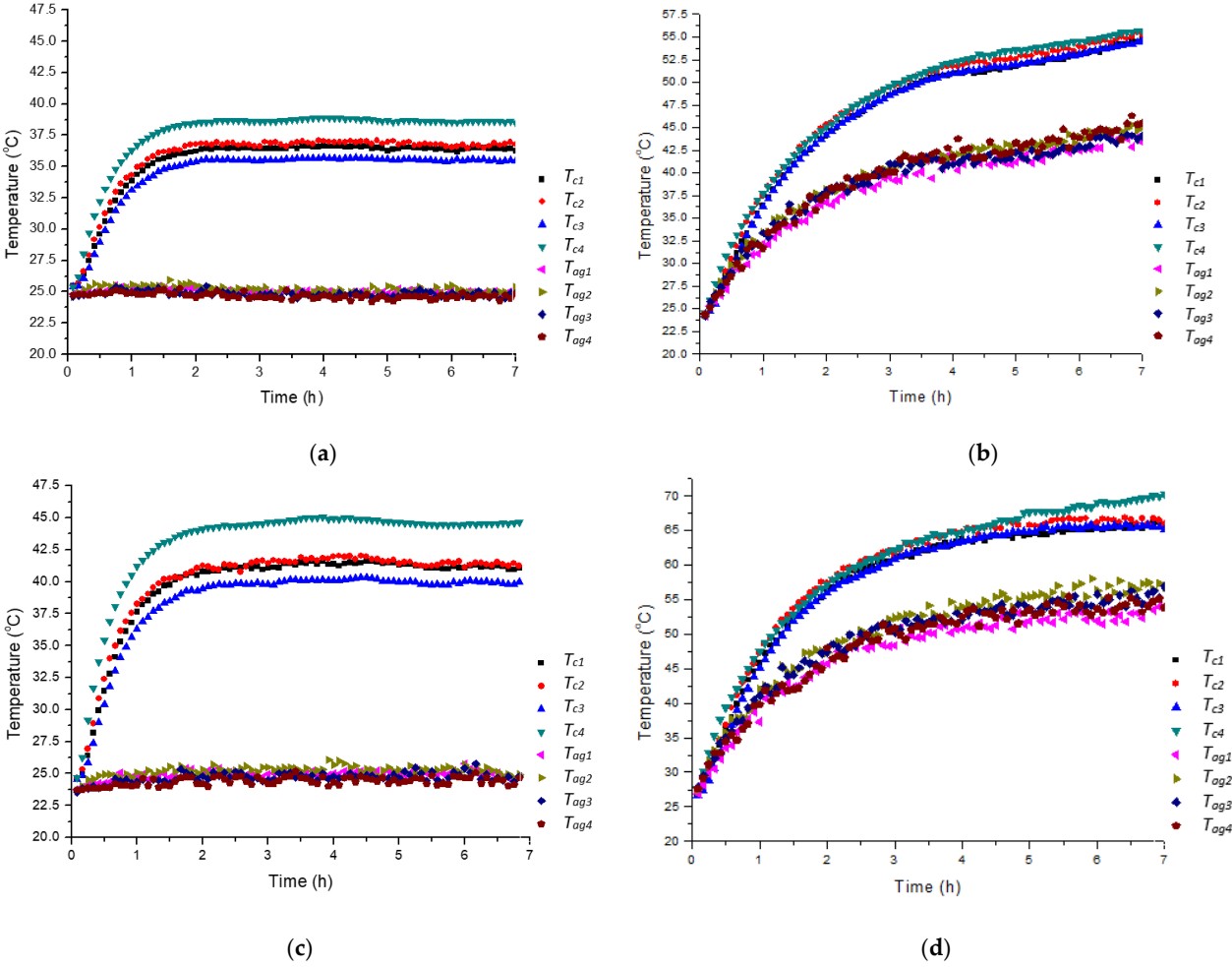

**Figure 8.** Temperature measurements with and without blockage of the vents at regulator voltages of 100 and 125 V. (**a**) 100 V open; (**b**) 100 V close; (**c**) 125 V open; (**d**) 125 V close.

From the graph above, it is clear that when the vents were closed, the temperatures on the canister surface, and especially in the air gap, increased significantly. At the steady state condition, the average temperatures of the canister surface ($T_{c1}$–$T_{c4}$) and the air gap ($T_{ag1}$–$T_{ag4}$) at a regulator voltage of 100 V increased from 36.88 to 52.97 °C and from 24.8 to 42.7 °C, respectively. While at 125 V, the average temperatures of ($T_{c1}$–$T_{c4}$) and ($T_{ag1}$–$T_{ag4}$) increased from 41.82 to 65.6 °C and from 24.9 to 54.33 °C, respectively.

To deepen the analysis, temperature data measurements were also carried out for eight half-closed vents, as shown in Figure 9.

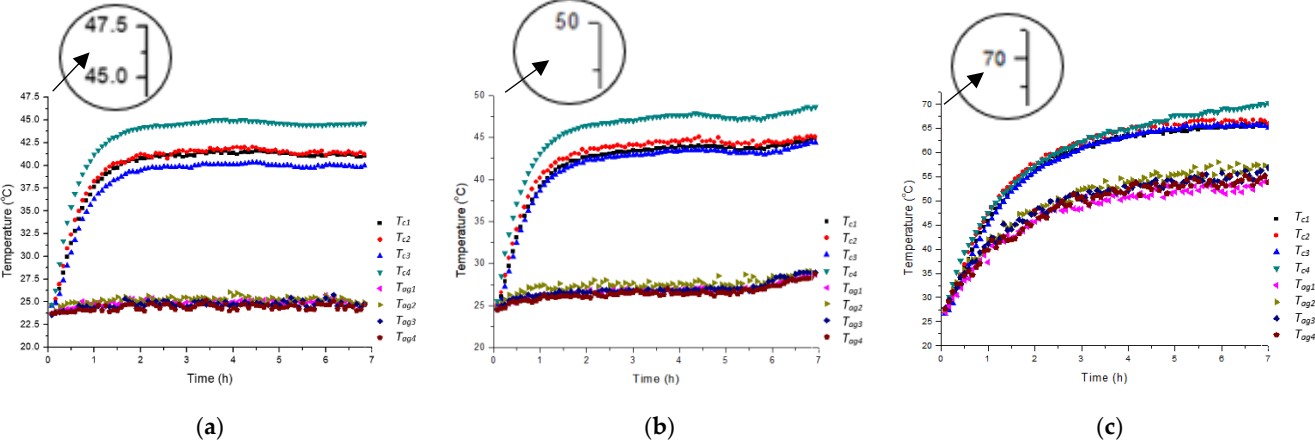

(a)     (b)     (c)

**Figure 9.** Temperature measurement for half-closed vents at a regulator voltage of 125 V. (**a**) 125 V open vents; (**b**) 125 V half-closed vents; (**c**) 125 V fully closed vents.

It can be seen in Figure 9 that the average canister temperature and the air gap temperature were higher than the average temperature for the fully open vents and lower than the fully closed vents.

As mentioned in Section 2.1, in addition to temperature data, air velocity measurements in the air gap were carried out. Figure 10 shows the air velocity data of the difference between the air temperature in the air gap and in the laboratory room. This temperature difference generates an air density difference, which causes airflow.

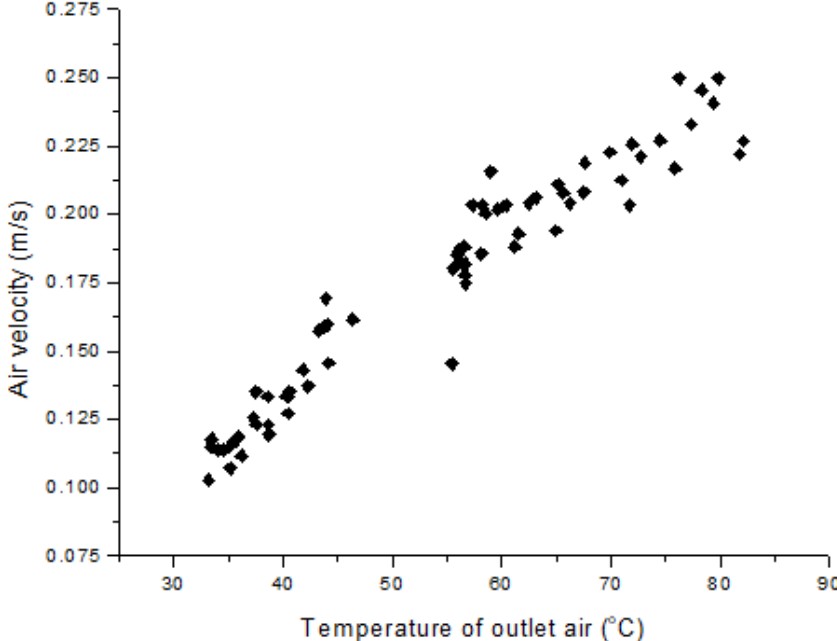

**Figure 10.** Air velocity versus air gap temperature.

*3.2. Measurement Data Comparison of Numerical Modeling, Simulation, and Theoretical Calculation*

3.2.1. Heat Transfer Analysis Using Software

The simulation and calculation analysis of the airflow in the air gap were carried out using CONTAM software (Figure 11).

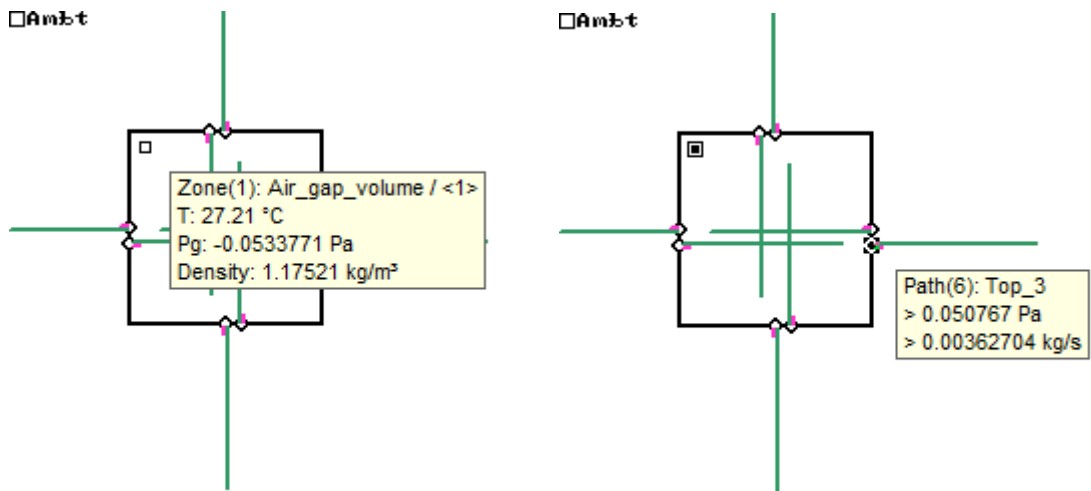

**Figure 11.** Airflow modeling using CONTAM software.

3.2.2. Heat Transfer Analysis Using Natural Air Mass Flow

The heat removal rate from the canister to the air gap was calculated using Equation (20), which calculates the natural airflow entering and leaving the dry storage vents through the air gap.

The velocity of the natural airflow in the air gap was obtained from the experimental data (Figures 6–9). These experimental velocity data were compared with the air velocity values from the calculation results of Equation (21) in addition to the values from the calculations using CONTAM software. The comparison results are presented in Figure 12.

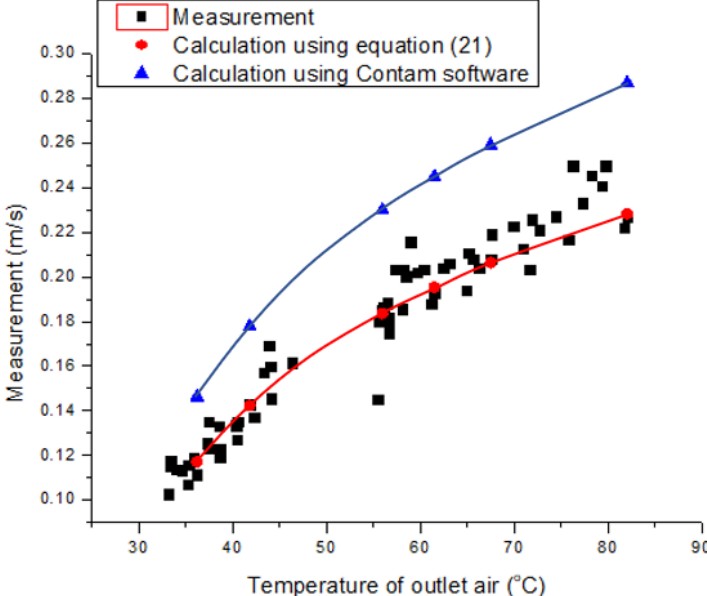

**Figure 12.** Air velocity in the air gap—experimental; manual calculation; software simulation.

From Figure 12, it is quite clear that the values from the manual calculation using Equation (21) (red diamond data) were quite close to the experimental data (black square data). The velocity data generated from the calculations using CONTAM (blue triangle

data) were higher (approximately 25.23%) than the manual calculations. The possible reason for this higher velocity value is that the CONTAM software did not consider the sharp turns of the air gap after passing through the inlet vents and at the outlet vents. Another possible reason is that the surface structure of the heater at the canister surface reduced the air velocity.

To obtain the value of the heat removal rate ($\dot{Q}_{rmv}$) from the airflow in the air gap, the air velocity in Equation (21) or the air volume flow in Equation (20) can be converted to air mass flow and then multiplied by the air-specific heat capacity and temperature difference between the outlet and inlet air flowing into the four bottom vents of the dry cask through the air gap and then flowing out from the four upper vents into the air-conditioned laboratory room. The heat removal rate equation is as follows:

$$\dot{Q}_{rmv} = C_d A \sqrt{(z_2 - z_1) g \frac{(T_{ag} - T_{lab\ room})}{T_{ag}}} \, \rho_{air} \, C_{v\ air} \left(T_{ag\ out} - T_{lab\ room}\right). \tag{22}$$

The specific heat capacity used in this study was not the isobaric heat capacity ($C_p$) but the isochoric heat capacity of air ($C_{v\ air}$) because the airflow in the air gap was caused by the difference between the pressure inside the air gap and outside the dry cask prototype. The air pressure difference is triggered by the air density difference, and the density difference is caused by the temperature difference.

The calculation results for $\dot{Q}_{rmv}$ using Equation (22) were compared with the calculation results using the heat transfer equation with the thermal resistance network, $\dot{Q}_{network}$ (discussed in Section 3.2.2), and also compared with the canister generation heat ($\dot{Q}_{can}$) values from the experiment. Some of the results of the comparison between the two calculated heat values, with the experimental results for 100 and 125 V of heater voltage, are shown in Table 1.

**Table 1.** Comparison between the experimental heat $\dot{Q}_{can}$, calculated heat $\dot{Q}_{rmv}$, and $\dot{Q}_{network}$.

| Heater Voltage | $\dot{Q}_{can}$ | $\dot{Q}_{rmv}$ | $\dot{Q}_{network}$ |
|---|---|---|---|
| 100 V open vents | 48 Watt | 42.70 Watt | 30.59 Watt |
| 125 V open vents | 75 Watt | 58.55 Watt | 45.01 Watt |
| 125 V half-closed vents | 75 Watt | 57.44 Watt | 48.29 Watt |

The experimental canister heat ($\dot{Q}_{can}$) was acquired from the measurement of the electric power or the electric current of the heater cable multiplied by the voltage ($P = V\,I$). The calculation results in Table 1 show that the $\dot{Q}_{can}$ values were higher than the calculated values, probably because less than 100% of the electric current at the heater was converted to heat energy. Another possible reason is that the radiation heat transfer from the canister to the air gap was not considered in the calculation.

### 3.2.3. Heat Transfer Analysis Using Thermal Resistance Network

As shown in Figure 4, the heat transfer from the spent fuel inside the canister to the outer wall of the dry cask prototype consisted of the heat transfer from the outer surface of the canister to the air gap, the heat transfer from the air gap through the cask wall, and the heat transfer from the outer surface of the cask wall to the air outside the dry storage prototype. Under the steady state condition, the heat transfer rate from the canister to the air gap was equal to the heat transfer rate from the air gap through the dry cask wall and also equal to the heat transfer rate from the wall to the air [22]. The heat transfer flow rate can be calculated using Equations (6)–(16).

In ventilated dry storage (the eight vents are open), the total thermal resistance can be calculated using Equation (7). If the eight vents are closed (non-ventilated dry storage), the total thermal resistance equation in this study becomes

$$R_{total} = R_{air\ conduction} + R_{cask\ wall} + R_{conv} = \frac{\ln\left(\frac{r2}{r1}\right)}{2\pi k_{air} h} + \frac{\ln\left(\frac{r3}{r2}\right)}{2\pi k h} + \frac{1}{h_5\ A_5} \tag{23}$$

where $k_{air}$ is the thermal conductivity of air in W/m.K.

In this case, the thermal resistance in the air gap was not calculated using the convection equation since there was no air movement because of closing all the vents. It was calculated using the conduction equation in a cylinder object.

The calculation results using Equation (23) and the Nusselt number for natural convection in a vertical cylinder using Equation (13) are shown in Table 1.

In Table 1, it can be seen that the calculation results using this thermal resistance network gave the lowest heat rate value compared to the experimental results and the calculation results using natural airflow (Equation (21)). This is probably because the average value of the air gap temperature ($T_{ag}$) measurement results used in this calculation was too low. In fact, the $T_{ag}$ value in the air gap was not uniform. The $T_{ag}$ values at the top of the canister were higher than the $T_{ag}$ values at the bottom of the canister. Another possibility is that the calculated $h_{conv}$ value was too low.

To quantify the accuracy of a modeling or calculation compared to some experimental data, the mean relative deviation (*MRD*) can be used. The *MRD* was used to measure the over- or under-prediction of the calculated values compared to the experiment values in this study. The *MRD* is defined as follows:

$$MRD = \frac{1}{N} \sum_{i=1}^{N} \frac{V_{pred} - V_{exp}}{V_{pred}} \tag{24}$$

where $V_{pred}$ and $V_{exp}$ are the predicted and experimentally determined $V$ values, respectively. $N$ is the data number.

In this study, the *MRD* value for the predicted air velocity values using Equation (21) compared to the experimental values was relatively very good, at +0.76%.

As mentioned above, the predicted air velocity was used to calculate the predicted heat removal of the canister heat.

The *MRD* values for heat rate prediction using natural airflow and the thermal resistance network equation were −23.69 and −29.54%. These large negative differences are due to the fact that the electrical energy in the heater was not 100% converted into heat and the radiation heat transfer from the canister to the air gap was not considered in the calculation. Another reason is the uncertainties of the measurements and the final results. The uncertainties of the final results (the combining uncertainty) can be calculated using partial differentiation method:

$$V = V\ (a,b) \tag{25}$$

$$\delta V = \frac{\partial V}{\partial a} \delta a + \frac{\partial V}{\partial b} \delta b \tag{26}$$

where δ is uncertainty.

Using Equation (26), high combining uncertainty values have been obtained. The uncertainties of the heat rate prediction using natural airflow and the thermal network are 33.03% and 24.8%, respectively.

## 4. Conclusions

The analysis results using airflow in the air gap in this study were proven to be close to the experimental results. The differences between the predicted and experimental values of air velocity in the air gap, the heat rate using the natural airflow equation, and the heat rate using thermal network resistance were relatively good (*MRD* = +0.76, −23.69,

and −29.54%, respectively). These two negative MRD values are quite high because the calculations carried out on these two values used several measurement data, each of which has uncertainties. Thus, high combining uncertainty values were obtained. In practice, the results of this study can be implemented as an alternative to and for the validation of the heat removal calculation using the convection heat transfer equation, which has been widely studied to calculate heat removal in ventilated dry storage.

This research can be developed by simulating the heat of all spent nuclear fuel elements stored in dry storage.

The possibility of a larger difference between the predicted value and the experimental results in other cases, for example, in dry storage with different dimensions, materials, and heat values, also needs to be investigated further.

The results of the study can be used to design dry storage in such a way that the heat removal will be optimal and the spent nuclear fuel stored in the dry storage will be kept at a lower temperature. In the long-term storage of spent fuel (more than 100 years), the spent fuel stored at lower temperatures will be safer from potential fuel cladding damage due to corrosion and material aging.

**Author Contributions:** Conceptualization, R.R.; methodology, R.R.; software, A.I. and A.A.R.; validation, R.R.; formal analysis, R.R.; investigation, R.R.; resources, A.A., K.H., P.A.A., Y.P., D.L.I.S., J.R. and R.S. (Risdiyana Setiawan); data curation, A.I. and R.R.; writing—original draft preparation, R.R.; writing—review and editing, R.R.; visualization, A.A.R.; supervision, R.R.; project administration, M.M. and W.W.; funding acquisition, R.S. (Raden Sumarbagiono). All authors have read and agreed to the published version of the manuscript.

**Funding:** This research was funded by the Ministry of Research and Higher Education under the PPTI 2021 Program, Grant No. 3/PPK/E4/2021.

**Institutional Review Board Statement:** Not applicable.

**Informed Consent Statement:** Not applicable.

**Data Availability Statement:** Not applicable.

**Acknowledgments:** We would like to thank the Research Centre for Radioactive Waste Technology, the Nuclear Power Research Organization (OR TN), the National Research and Innovation Agency (BRIN) for supporting this research.

**Conflicts of Interest:** The authors declare no conflict of interest.

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
