# Peer review of "Theoretical and Experimental Analysis on Influence of Natural Airflow on Spent Fuel Heat Removal in Dry Cask Storage"

_sustainability, doi:10.3390/su14031859_

Round 1

Reviewer 1 Report

comments to the paper are provided in the text

Reviewer 2 Report

See attached file

Reviewer 3 Report

The manuscript needs rewriting and editing. 

The following comments are provided for reference.

(1) The purpose of the study is not clear. Why the heat removal needs theoretical prediction, simulation and experimental validation?

(2) Is the studied cask system a real case? or just a model?  More details needs about the studied cask system.

(3) In Fig. 12 and Table 1,the theoretically predicted data, software simulated data, and experimental data were compared. The difference among theoretical prediction, simulation, and experiment needs explaining in more detail.

(4) MRD (mean relative deviation) is not commonly used. More explanations are needed.  

(5)   From the results and discussion, no specific concluding remarks can be made about the natural convection and heat removal of a cask system.  

Round 2

Reviewer 1 Report

Paper looks good and could be published. Thanks to authors for the conducted works.

Author Response

Dear Reviewer,

thank you very much for your comments and suggestions that allowed us to improve the quality of our manuscript.

Reviewer 2 Report

General comments

  1. English is improved but authors still need to read the text carefully to correct terminology.
  2. It is necessary to provide relative uncertainties of the measurements and of the final results and then it will be possible to see how big they are. It seems they are really big!
  3. Authors have not demonstrated that they are experienced to perform modeling using ANSYS/FLUENT. Therefore, it is proposed to delete all the information related to this modeling from the paper.
  4. After updating the text, it is necessary to update the conclusions also.

Detailed comments are provided in the attached file.

Author Response

Dear Reviewer 2,

Grateful for your second review that allowed us to further improve our manuscript.

herewith we attached the revised manuscript and the detailed response to reviewer's comments (at the last page of the document)

thank you very much.

Reviewer 3 Report

It is very difficult to review the manuscript in present format.  

The manuscript is full of the remarks made for English editing.

The authors should add the page number for each of the responses to reviewers' comments.  

There are some comments that the reviewer would recommend the authors to refine the manuscript.

  1. As described in the section of Abstract, “…The theoretical analysis results in this study are were relatively consistent with the experimental 24 results, with the MRD (mean relative deviation) values of +0.76%, -23.69 % and -29.54%.” (Lines 24-25).

There should be only one MRD value for the ratio of theoretical analysis results to experimental results. But three MRD values were listed there.  

  1. As described in the section of introduction, “…There are several studies on the handling and storage on of spent fuel [1][2][3][4][5][6][7][8][9][10]…” (Lines 33-34)

The citation format should be corrected to [1-10].

“ Several studies on heat removal in from non-ventilated dry storage, among others, have discussed decay heat decay heat removal by after a long-term storage period using the ALGOR code. [11][12],” [Lines 50-52]

The citation format should be corrected to [11, 12].

“Regarding the ventilated dry storage, there are more studies that have discussed 55 heat removal [15][16][17][18]…” [Lines 55-56]

The citation format should be corrected to [15-18].

“In this study, the spent fuel that will be stored in the dry storage is a spent fuel from our research reactor (materials testing reactor/MTR), with a fuel temperature limit lower than 150 °ËšC  [19][20].” [Lines 79-81]

The citation format should be corrected to [19, 20]

  1. “…we have did not found find any study that focused on the an analysis on of the numerical prediction of the natural airflow in the air gap…” [Lines 71-73 ]

To the reviewer’s knowledge, the natural airflow in the air gap has been studied (e.g. [15]). The authors cannot claim in that way.    

  1. In the section of 3.2.1 Heat transfer analysis using software, a brief introduction should be added to explain the Fluent and CONTAM software. To compare with experimental data, the authors should set the same boundary condition and analysis parameters in the simulation or prediction. Detailed explanations should be added.

  1. In Fig. 14, the air velocity calculated using Fluent software is totally different from the others. It means that the calculation using Fluent software is not compatible with the others. Therefore it is recommended to remove all the results relevant to Fluent.

  1. “The MRD values for heat rate prediction using natural air flow and using thermal resistance network equation are were -23.69% and -29.54%.” (Lines 376-377)

By the definition of MRD, the negative value means that the prediction may underestimate the experiment or true value. It is not good for engineering design and applications. The authors therefore should comment this and suggest for further study.

Author Response

Dear Reviewer 3,

Grateful for your second review that allowed us to further improve our manuscript.

herewith we attached the revised manuscript and the detailed response to reviewer's comments (at the last page of the document)

thank you very much.

Round 3

Reviewer 2 Report

It is positive that information related with the modeling using FLUENT code is deleted.

Some relative uncertainties are provided. They are really large, especially having in mind that investigations are oriented to the nuclear industry.